# Time-Course Transcriptome Analysis Reveals Distinct Phases and Identifies Two Key Genes during Severe Fever with Thrombocytopenia Syndrome Virus Infection in PMA-Induced THP-1 Cells

**DOI:** 10.3390/v16010059

**Published:** 2023-12-29

**Authors:** Tao Huang, Xueqi Wang, Yuqian Mi, Wei Wu, Xiao Xu, Chuan Li, Yanhan Wen, Boyang Li, Yang Li, Lina Sun, Jiandong Li, Mengxuan Wang, Tiezhu Liu, Shiwen Wang, Mifang Liang

**Affiliations:** 1State Key Laboratory for Molecular Virology and Genetic Engineering, National Institute for Viral Disease Control and Prevention, Chinese Center for Disease Control and Prevention, Beijing 102206, China; thomas_ht@163.com (T.H.); wuwei@ivdc.chinacdc.cn (W.W.); xxgenevax@sina.com (X.X.);; 2Capital Institute of Pediatrics, Beijing 100020, China; iwangxueqi@gmail.com; 3Shanxi Academy of Advanced Research and Innovation, Taiyuan 030032, China; mi_yq@saari.org.cn; 4Chongqing Research Institute of Big Data, Peking University, Chongqing 400039, China

**Keywords:** THP-1, phorboll-12-myriate-13-acetate, SFTSV infection, multiple time points

## Abstract

In recent years, there have been significant advancements in the research of Severe Fever with Thrombocytopenia Syndrome Virus (SFTSV). However, several limitations and challenges still exist. For instance, researchers face constraints regarding experimental conditions and the feasibility of sample acquisition for studying SFTSV. To enhance the quality and comprehensiveness of SFTSV research, we opted to employ PMA-induced THP-1 cells as a model for SFTSV infection. Multiple time points of SFTSV infection were designed to capture the dynamic nature of the virus–host interaction. Through a comprehensive analysis utilizing various bioinformatics approaches, including diverse clustering methods, MUfzz analysis, and LASSO/Cox machine learning, we performed dynamic analysis and identified key genes associated with SFTSV infection at the host cell transcriptomic level. Notably, successful clustering was achieved for samples infected at different time points, leading to the identification of two important genes, PHGDH and NLRP12. And these findings may provide valuable insights into the pathogenesis of SFTSV and contribute to our understanding of host–virus interactions.

## 1. Introduction

Severe Fever with Thrombocytopenia Syndrome Virus (SFTSV) is a highly pathogenic pathogen belonging to the Bunyavirales order. It can be transmitted through tick bites [1] as well as person-to-person contact [2,3,4] and exposure to the blood of infected individuals [5,6]. Additionally, there have been reports that some veterinarians and pet owners in Japan have been infected with SFTSV from SFTS animals [7]. Clinical manifestations of SFTSV infection include fever, thrombocytopenia, and gastrointestinal reactions, with severe cases leading to multiple organ failure and death [1]. Despite global attention, the pathogenic mechanism of SFTSV infection remains unclear, and there is a lack of effective vaccines and treatments. Additionally, our understanding of the molecular mechanisms and immune response regulation in SFTSV infection is limited. On one hand, the basic biological characteristics and infection mechanism of SFTSV, a newly discovered virus, have not been fully elucidated, particularly in the human host [8]. Furthermore, there is a relative lack of research on the molecular mechanism of SFTSV infection and the regulation of immune response. On the other hand, due to the high pathogenicity of SFTSV, laboratory research on this virus carries inherent risks, which restricts researchers from conducting in-depth investigations into its infection mechanism. Therefore, it is of paramount importance not only to prevent and control the spread of SFTSV but also to urgently conduct further comprehensive research on its infection and pathogenesis.

The THP-1 cell line, derived from patients with acute monocytic leukemia, is widely used in immunology, viral infection, and cell biology research [9]. Its ability to differentiate into mononuclear/macrophage cells makes it an ideal model for studying mononuclear/macrophage function, mechanisms, and signaling pathways [10,11]. Compared to human peripheral blood mononuclear cells (PBMCs), THP-1 cells are easier to culture, have a stable genetic background, and exhibit less individual variation, making them suitable for studying viral infection and immune response [12,13]. Thus, investigating SFTSV infection in THP-1 cells can provide valuable insights into the interaction between the virus and human host cells as well as the mechanisms and pathogenesis of SFTSV infection.

Transcriptomics, a research method for studying gene expression, can reveal comprehensive gene expression changes and regulatory mechanisms [14]. In the context of SFTSV, transcriptomic analysis can uncover the overall expression and changes in host transcription during infection, identify genes and pathways related to viral infection, and shed light on the interaction between SFTSV and host cells as well as the infection and pathogenesis of SFTSV.

Therefore, this study aims to perform transcriptomic analysis on THP-1 cells infected with SFTSV at different time points. By examining the changes in host transcription and immune response regulation during SFTSV infection, we can gain insights into the virus–host interaction and the underlying mechanisms. This research will provide a theoretical basis for preventing and controlling SFTSV infection as well as novel ideas and approaches for the clinical treatment of SFTS. Furthermore, the findings from this study will contribute to a deeper understanding of the molecular mechanisms and immune response regulation in SFTSV infection, paving the way for future research and offering new perspectives for the clinical management of related diseases.

## 2. Materials and Methods

### 2.1. Cells, Viruses, Antibodies and Other Reagents

The THP-1 cells were obtained from the ATCC cell line, and the virus was a strain HB29 of the SFTSV virus (GenBank No. HM745930, HM745931, HM745932) isolated from diagnosed patients with SFTS by the Chinese Center for Disease Control and Prevention. The antibodies used in this study included Mouse Anti-Human CD14 (FITC marker) (BD) (Franklin Lakes, NJ, USA). Additionally, commonly used reagents and materials included 4% paraformaldehyde (Biosharp) (Hefei, China), Triton X-100 (Beyotime) (Shanghai, China), Saponin (Beyotime), RPMI 1640 (ThermoFisher) (Waltham, MA, USA), EDTA (Gibco) (Grand Island, NY, USA), BSA (Coolaber) (Beijing, China), fetal bovine serum (FBS) (Gibco) (Grand Island, NY, USA), Penicillin/Streptomycin (PS) (Gibco) (Grand Island, NY, USA), phorboll-12-myriate-13-acetate(PMA) (Sigma Aldrich) (St. Louis, MO, USA), DAPI staining solution (Beyotime) (Shanghai, China), 8-well chamber cover glass (ThermoFisher) (Waltham, MA, USA), etc. The detailed procedural steps of the experiment are illustrated in Figure 1A.

### 2.2. Induction of THP-1

THP-1 cells were maintained in RPMI 1640 with 10% FBS and 1% PS added at 37 °C and 5% CO_2_. In separate infection experiments, monocyte activation was induced by stimulation of cells with 80 ng/mL of phorboll-12-myriate-13-acetate (PMA) for 24 h [15,16].

### 2.3. Cell Samples Preparation for Flow Cytometry

The induced THP-1 cells, converted to an adherent form, were digested with EDTA and collected by centrifugation. The cells were then resuspended in PBS with 1% FBS and subjected to another round of centrifugation. Afterward, the cell pellet was stained with a mixture of staining buffer and CD14 FITC antibody, followed by incubation in a light-free environment. The cells were then cleaned and collected through multiple rounds of centrifugation. Finally, the cells were resuspended in PBS with 1% FBS and loaded onto a flow cytometry tube for analysis [17].The proportion of THP-1 cells showing FITC fluorescence was calculated as the induction rate of PMA-induced THP-1 cells.

### 2.4. Cell Samples Preparation for Cellular Immunofluorescence

THP-1 cells induced by PMA and THP-1 cells treated with RPMI 1640 were cultured on 8-well glass slides. At 60% density, cells were fixed with 4% paraformaldehyde for 30 min at room temperature. After rinsing with 1 × PBS, the cell membrane was sealed with 0.15% Saponin solution. Following removal of the sealing solution, cells were incubated with PBS containing 3% BSA. CD14 FITC antibody was added and incubated in the dark or covered with tinfoil. After rinsing with PBS, DAPI solution was applied for nuclear staining. Cells were observed using fluorescence microscopy.

### 2.5. SFTSV Infection in dTHP-1 Cells

PMA-induced THP-1 cells were batch-infected with SFTSV(HB29). For the purpose of viral infection, THP-1 macrophages were allowed to reach 80–90% confluence before infection with SFTSV at a specified multiplicity of infection (MOI = 0.5) using RPMI medium 1640 containing 2% FBS for 2 h. The infection process was carried out under optimal conditions to ensure maximum infectivity and minimal cell death. After removing the supernatant, cells were incubated with standard SFTSV for 1 h at 37 °C. Following virus removal and PBS wash, cells were incubated in cell preservation solution. Different time points (0.5 h, 2 h, 8 h, 24 h, and 48 h) were collected by EDTA digestion and centrifugation. Cell pellets were resuspended, cleaned, and stored for further experiments.

### 2.6. Bioinformatics Analysis

#### 2.6.1. Principal Component Analysis

We conducted principal component analysis (PCA) using the R software package stats (version 3.6.0). Firstly, missing values in the expression profile were addressed by removing rows or columns with more than 50% missing data. The expression profile was then normalized through log2 (x + 1) transformation. Next, z-score normalization was applied to the expression profile. Finally, we used the prcomp function for dimensionality reduction analysis, resulting in the final reduced matrix. All statistical analyses were performed using R software (Version R-4.3.1 for Windows). Significance was determined at *p* < 0.05 or *p* < 0.01. The detailed bioinformatics analysis workflow is outlined in Figure 1B.

#### 2.6.2. DEGs Identification

Linear models for microarray data (Limmas) are a differential representation screening method based on generalized linear models [18]. The differential expression analysis was performed using the limma (version 3.40.6) package in R. The lmFit function was used for multiple linear regression of the expression profiles. The eBays function was used to compute moderated t-statistics, moderated F-statistics, and log-odds of differential expression through empirical Bayes moderation of standard errors. Differential genes were identified based on a fold change threshold of 1.5 and a false discovery rate (FDR) < 0.05. Volcano plots and heat maps were used for visualization.

#### 2.6.3. Function and Pathway Enrichment Analysis

For gene function and pathway enrichment analysis, we utilized the KEGG REST API (https://www.kegg.jp/kegg/rest/keggapi.html (accessed on 1 October 2023)) to obtain the latest gene annotation of pathways. The clusterProfiler (version 3.14.3) package in R was employed for enrichment analysis [19]. Enriched gene sets with a *p*-value < 0.05 and an FDR < 0.25 were considered statistically significant.

#### 2.6.4. WGCNA Analysis

WGCNA (Weighted Gene Co-expression Network Analysis) was performed to construct a scale-free co-expression network. MAD (Median Absolute Deviation) was calculated for each gene, and outlier genes and samples were removed [20]. Pearson’s correlation matrices and average linkage hierarchical clustering were utilized to construct a weighted adjacency matrix. The power parameter was set to 9. The topological overlap matrix (TOM) was derived from the adjacency matrix, and dissimilarity (1-TOM) was calculated. Genes with similar expression profiles were classified into modules using average linkage hierarchical clustering with a minimum module size of 30. Sensitivity was set to 3.

#### 2.6.5. LASSO/Cox Regression Analysis

Survival time, survival state, and gene expression data were integrated using the glmnet package in R, and LASSO/Cox regression analysis was performed [21].

#### 2.6.6. Mfuzz Soft Cluster Analysis

The Mfuzz package in R was used for soft clustering analysis of microarray data. Fuzzy C-Means clustering (FCM) was employed to analyze the time trend of gene expression changes, and genes with similar RNA expression patterns were classified into clusters [22].

#### 2.6.7. GSEA Results Display

Gene Set Enrichment Analysis (GSEA) was conducted using the GSEA software (version 3.0) [23]. Samples were divided into high and low expression groups based on the expression level of the gene. The Molecular Signatures Database was utilized for pathway analysis [24], and the top ten KEGG pathways with positive and negative correlations were visualized. A significance threshold of *p* < 0.05 and FDR < 0.25 was applied.

#### 2.6.8. GEO Datasets Involved in This Study

The first dataset used in this study comes from GSE144358 (platform: GPL20795 HiSeq X Ten (Homo sapiens)). It was retrieved from the Gene Expression Omnibus (GEO) database (http://www.ncbi.nlm) accessed on 1 November 2023. This dataset includes 21 samples from healthy humans, 19 samples from recovered humans, 22 samples from acute recovered humans, and 15 samples from acute deceased humans.

To assess the specificity of the identified feature genes, this study conducted an analysis of four additional GEO datasets (Accession IDs: GSE18816 [25], GSE41300 [26], GSE141236 [27], GSE162261 [28]). These datasets were utilized as external validation sets to examine the expression patterns of the target feature genes in both clinical samples and macrophages cultured in the laboratory. Specifically, the GSE18816 dataset comprises 9 samples infected with H1N1, 9 samples infected with H5N1, and 9 mock control samples. The GSE41300 dataset consists of 9 samples infected with Lassa Virus, 9 samples infected with Mop/Las virus, and 9 uninfected control samples. In the GSE141236 dataset, there are 3 samples of macrophages infected with HCMV and 3 uninfected control macrophages. Lastly, the GSE162261 dataset includes 3 samples infected with IAV and 3 uninfected control samples. These diverse datasets serve as crucial external benchmarks for evaluating the robustness and reliability of the identified feature genes in different experimental conditions.

## 3. Results

### 3.1. PMA Induction Results of THP-1 Cells

Flow cytometry analysis revealed significant differences between the PMA-induced dTHP-1 cells and the control THP-1 cells. As shown in the flow cytometry sorting results figure, Figure 2A,B,F,G represent the preliminary steps of gradually screening the target cell population of PMA-induced dTHP-1 cells and control THP-1 cells samples, respectively. Finally, Figure 2C,D,H,I were obtained, which can be clearly observed through the corresponding statistical results (Figure 2E,J). The proportion of CD14-positive cells, as detected by flow cytometry using CD14-FITC antibody, was 75.4% in the PMA-induced dTHP-1 cells (Figure 2D,E) compared to only 3.1% in the control cells without PMA induction (Figure 2I,J). The results showed that under the conditions of 24 h PMA treatment at 80 ng/mL, the proportion of THP-1 cells induced to differentiate into macrophages was approximately 75.4%. These results are consistent with previous studies [15] and meet the requirements for subsequent SFTSV infection experiments.

### 3.2. Fluorescence Microscopy Results of PMA-Induced and Normal Control THP-1 Cells

The fluorescence microscopy images (Figure 3) demonstrate the staining patterns of dTHP-1 cells induced by PMA and the control THP-1 cells. Figure 3A shows the DAPI staining of dTHP-1 cells, Figure 3B shows the FITC staining of dTHP-1 cells, and Figure 3C presents the merged result of DAPI and FITC staining for dTHP-1 cells. Similarly, Figure 3D–F display the corresponding staining results for the control THP-1 cells. The results indicate a significant increase in the proportion of dTHP-1 cells in the THP-1 cell population after PMA induction, consistent with the findings from flow cytometry analysis.

After establishing the treatment conditions to induce dTHP-1 cells from THP-1 by PMA, we conducted batch culture and induction of THP-1 cells. Cell samples were collected at different time points (control, 0.5 h, 2 h, 8 h, 24 h, and 48 h) during SFTSV infection of dTHP-1 (Figure 1A). Each time point was performed in triplicate, resulting in a total of 18 cell samples, including the control group, for subsequent transcriptome sequencing.

### 3.3. Transcriptome Sequencing Data Quality Control

The error rate for base calling (Error (%)) did not exceed 0.05%, meeting the standard requirements for data quality control in transcriptome sequencing. Moreover, the distribution of sequencing error rates demonstrated that Q20 (%) and Q30 (%) values, representing the percentage of bases with quality scores greater than 20 and 30, respectively, exceeded 97% and 92%, respectively, in all samples. These high-quality scores indicate reliable base identification and minimal chances of base misinterpretation. Additionally, the GC content distribution in the sequenced samples did not show any significant deviation between AT and GC bases, indicating that the sequencing and library preparation processes had minimal impact on the sequencing results. Overall, the sequencing data in this study passed rigorous quality evaluation and can be confidently used for subsequent statistical analysis (Table 1).

### 3.4. Differential Analysis between SFTSV-Infected THP-1 Cells and the Control Group

Principal component analysis (PCA) revealed significant differences in distribution between SFTSV-infected dTHP-1 cells and the control group (Figure 4A). The heatmap analysis of the top 100 differentially expressed genes (50 upregulated and 50 downregulated) clearly distinguished the infected cells from the control group (Figure 4B). Additionally, the volcano plot analysis identified a total of 5049 differentially expressed genes, with 4072 upregulated and 977 downregulated genes, meeting the criteria of a fold change greater than 1.5 and FDR < 0.05 (Figure 4C). These results provide valuable insights into the molecular alterations associated with SFTSV infection in dTHP-1 cells.

Differential gene analysis was performed between SFTSV-infected THP-1 cells and the control group. To gain deeper insights into these differentially expressed genes, we applied the WGCNA method for further analysis. The soft threshold power (β) of 9, with a scale-free topology index (R^2^) of 0.79 (Figure 5A), was chosen to construct the adjacency matrix in order to ensure a scale-free network. The average connectivity value of 400.87 (Figure 5B) was selected to strike a balance between module preservation and granularity, where higher values result in larger modules with fewer distinct functional themes and lower values lead to smaller, more specific modules. Through module merging based on dissimilarity calculations, 10 co-expression modules were obtained (Figure 5C). In addition, we conducted a comparative analysis of the correlation between different modules (Figure 5D). Notably, the darkviolet module showed the highest correlation (0.78) with the infection group (Figure 5E). By assessing gene significance (GS) and module membership (MM) [29], we identified 515 hub genes with high connectivity within the clinically significant module, using a cutoff criteria of |MM| > 0.8 and |GS| > 0.1. We performed a comprehensive analysis to determine the associations between modules and specific traits, revealing a significant correlation between the darkviolet module and SFTSV infection samples (Figure 5E). These findings shed light on key factors distinguishing SFTSV infection from the control group.

Furthermore, the clustering analysis of the top 100 differentially expressed genes in the heatmap revealed distinct clustering patterns among the dTHP-1 cell samples infected with SFTSV at different time points (Figure 4B). To gain a deeper understanding of the correlation and clustering results among samples at various time points (control, 0.5 h, 2 h, 8 h, 24 h, and 48 h) of SFTSV infection in dTHP-1 cells, we performed correlation heatmap and clustering tree analyses (Figure 6A,B). The findings indicated that the samples from the control group and SFTSV-infected groups at 0.5 h–2 h, 8 h, 24 h, and 48 h formed distinct clusters, displaying substantial differences in their correlation patterns (Figure 6A,B).Therefore, we conducted LASSO/Cox regression analysis on the expression profiles of the 515 hub genes obtained from WGCNA analysis. The analysis was performed separately for different time points of SFTSV infection and the control group. As a result, we identified distinct sets of characteristic genes that effectively differentiate each time point from the control group (Figure 6C–F).

We conducted UpSet plot intersection analysis on these key genes. Interestingly, within these intersections, two genes, PHGDH and NLRP12, consistently appeared at all time points (Figure 7A and Appendix A), and their expression trends remained largely consistent across various infection time points (Figure 7B). Subsequently, we performed Gene Set Enrichment Analysis (GSEA) on these two characteristic genes. The results revealed that PHGDH was enriched in the top-10-ranked signaling pathways, including UBIQUITIN MEDIATED PROTEOLYSIS, REGULATION OF ACTIN CYTOSKELETON, TGF BETA SIGNALING PATHWAY, LYSINE DEGRADATION, T-CELL RECEPTOR SIGNALING PATHWAY, DRUG METABOLISM CYTOCHROME P450, LONG-TERM POTENTIATION, PROXIMAL TUBULE BICARBONATE RECLAMATION, B-CELL RECEPTOR SIGNALING PATHWAY, and PROGESTERONE-MEDIATED OOCYTE MATURATION. On the other hand, NLRP12 was enriched in the top-10-ranked signaling pathways, including CHEMOKINE SIGNALING PATHWAY, FOLATE BIOSYNTHESIS, TOLL-LIKE RECEPTOR SIGNALING PATHWAY, CYTOSOLIC DNA SENSING PATHWAY, RIG-I-LIKE RECEPTOR SIGNALING PATHWAY, REGULATION OF AUTOPHAGY, CYTOKINE–CYTOKINE RECEPTOR INTERACTION, FC EPSILON RI SIGNALING PATHWAY, PRIMARY BILE ACID BIOSYNTHESIS, and CALCIUM SIGNALING PATHWAY (Figure 7C,D).

Based on the enrichment of these two genes, PHGDH and NLRP12, in multiple significant signaling pathways, it can be inferred that they play important roles in various cellular processes. PHGDH’s enrichment in pathways such as UBIQUITIN MEDIATED PROTEOLYSIS, TGF BETA SIGNALING PATHWAY, and DRUG METABOLISM CYTOCHROME P450 suggests its involvement in protein degradation, cellular response to growth factors, and drug metabolism. These pathways are essential for maintaining cellular homeostasis and regulating various physiological functions. NLRP12’s enrichment in pathways like CHEMOKINE SIGNALING PATHWAY, TOLL LIKE RECEPTOR SIGNALING PATHWAY, and RIG I LIKE RECEPTOR SIGNALING PATHWAY indicates its potential role in immune response and inflammation regulation. These pathways are crucial for detecting pathogen-associated molecular patterns (PAMPs) and activating immune responses against infectious agents. Overall, these findings suggest that PHGDH and NLRP12 may have significant implications in SFTS infection and its related biological processes, emphasizing their potential importance in disease development and progression.

In addition, we performed fuzzy clustering (MUfzz) expression trend analysis on the expression profiles of the 515 hub genes obtained from WGCNA analysis at different time points of SFTSV infection (Figure 8A–I). In Figure 8, we present the fuzzy clustering (MUfzz) analysis conducted on the expression profiles of 515 hub genes identified through WGCNA analysis at various time points during SFTSV infection. The time series pattern clustering analysis in Mfuzz is a visual representation of fuzzy clustering, showcasing the fuzzy membership relationships of each sample to different clusters. The color intensity of the lines may indicate the degree of membership, with darker colors suggesting higher membership levels. This visualization method serves to illustrate the ambiguous associations of each sample with different clusters in a given time series context. The detailed description and visual representation provided in Figure 8 offer valuable insights into the expression trends of the identified hub genes at different time points of SFTSV infection, as determined through fuzzy clustering analysis. Interestingly, the expression trend of cluster 4 was consistent with the transcriptional changes observed in our identified characteristic genes, PHGDH and NLRP12 (Figure 8D).

Therefore, we speculate that there may be other important genes within the gene set associated with cluster 4. As a result, we conducted heatmap analysis on the expression profiles of 50 genes within the cluster 4 gene set (Figure 9A), and intriguingly, both PHGDH and NLRP12 were indeed present in this gene set associated with cluster 4.

To further investigate the overall functional profiles of these 50 genes, we conducted GO and KEGG functional enrichment analysis. The results revealed that these genes were enriched in various biological processes (BPs), cellular components (CCs), and molecular functions (MFs). The top enriched BP terms included regulation of ventricular cardiac muscle cell membrane repolarization, regulation of cardiac muscle cell membrane repolarization, ventricular cardiac muscle cell membrane repolarization, regulation of membrane repolarization, ventricular cardiac muscle cell action potential, cardiac muscle cell membrane repolarization, L-serine biosynthetic process, membrane repolarization, L-serine metabolic process, and cardiac muscle cell action potential involved in contraction. The top enriched CC terms included syntrophin complex, BRISC complex, G protein-coupled receptor dimeric complex, G protein-coupled receptor complex, membrane raft, membrane microdomain, membrane region, anchored component of external side of plasma membrane, dystrophin-associated glycoprotein complex, and glycoprotein complex. The top enriched MF terms included ion channel binding, modification-dependent protein binding, calmodulin binding, phosphorylation-dependent protein binding, methylated histone binding, methylation-dependent protein binding, microtubule-severing ATPase activity, beta-adrenergic receptor kinase activity, voltage-gated potassium channel activity involved in atrial cardiac muscle cell action potential repolarization, and hydrolase activity acting on carbon–nitrogen (but not peptide) bonds in cyclic amides. In summary, the top 10 enriched pathways in GO analysis were related to the regulation of ventricular cardiac muscle cell membrane repolarization, regulation of cardiac muscle cell membrane repolarization, ventricular cardiac muscle cell membrane repolarization, regulation of membrane repolarization, ventricular cardiac muscle cell action potential, cardiac muscle cell membrane repolarization, L-serine biosynthetic process, membrane repolarization, L-serine metabolic process, and cardiac muscle cell action potential involved in contraction. In KEGG analysis, the top 10 enriched functional signaling pathways included glycine, serine, and threonine metabolism; acute myeloid leukemia; biosynthesis of amino acids; EGFR tyrosine kinase inhibitor resistance; ErbB signaling pathway; longevity regulating pathway, choline metabolism in cancer; HIF-1 signaling pathway; cholinergic synapse; and glutamatergic synapses.

Based on the provided information, it is important to note that the functional enrichment results of these 50 genes are related to the context of studying differential gene expression in THP-1 cells infected with the SFTSV. The enrichment of genes involved in the regulation of ventricular cardiac muscle cell membrane repolarization, cardiac muscle cell membrane repolarization, and L-serine biosynthetic/metabolic processes may indicate potential implications for cardiovascular function in the context of viral infection. Based on the results of the enrichment function that the repolarization of the myocardium and the disturbance of ion channel activity can lead to cardiac dysfunction, we hypothesized that viral infection may also affect the function of normal cardiac tissue. The presence of enriched components such as syntrophin complex, G protein-coupled receptor complexes, and membrane rafts/microdomains suggests the possible involvement of these genes in signaling pathways and cellular communication processes during viral infection. These components play crucial roles in cellular responses, including immune signaling, which are likely relevant in the context of viral pathogenesis and host defense mechanisms. Furthermore, the molecular functions exhibited by these genes, such as ion channel binding, protein interactions, and enzyme activities, imply their involvement in modulating cellular processes during viral infection. For example, ion channel binding may indicate a role in regulating ion fluxes, which could be critical for viral replication or host immune responses. In terms of the enriched pathways, several notable associations can be made. The glycine, serine, and threonine metabolism pathway suggests a potential link between these genes and amino acid metabolism, possibly indicating alterations in metabolic pathways during viral infection. The acute myeloid leukemia pathway enrichment might suggest a connection between these genes and hematopoietic disorders, which could be relevant considering the impact of viral infections on immune cell development and function. Overall, these functional enrichments provide insights into the potential roles of these 50 genes in the context of Novel Bunyavirus infection in THP-1 cells. Further investigations into the specific functions and interactions of these genes may contribute to a better understanding of the underlying mechanisms involved in viral pathogenesis, host response, and potential therapeutic targets for mitigating viral-infection-associated complications.

Furthermore, in order to gain further insights into the potential clinical relevance of these two genes, we explored the transcriptional expression profiles of patients with varying degrees of severity in SFTS through the GEO database (GSE144358) [30]. The results revealed the expression levels and trends of PHGDH and NLRP12 genes. The analysis demonstrated that the expression level of the PHGDH gene exhibited a linear increase with the worsening of disease severity, showing significantly higher expression in SFTS patients who experienced severe outcomes leading to fatality (Figure 10A). On the other hand, although the trend was not as pronounced as observed for PHGDH, NLRP12 also showed significant differential expression across different disease severities (*p* < 0.05) (Figure 10B). These findings suggest that PHGDH may play a crucial role in the progression and prognosis of SFTS, particularly in severe cases, while NLRP12 may also contribute to disease severity. Further investigations are warranted to elucidate the precise mechanisms by which these genes impact the pathogenesis and clinical outcomes of SFTS.

To further assess the specificity of the feature genes PHGDH and NLRP12, this study analyzed four additional GEO datasets (Accession IDs: GSE18816, GSE41300, GSE141236, GSE162261) as external validation sets. These datasets were utilized to examine the expression patterns of the target feature genes in clinical samples and laboratory-cultured macrophages. Two of these datasets involve clinical samples infected with viruses that induce immune dysregulation, while the other two datasets pertain to infections conducted on macrophages cultured in the laboratory using viruses. Results indicate that there is no significant difference in the expression of the feature genes PHGDH (*p* = 0.94) (Figure 11A) and NLRP12 (*p* = 0.84) (Figure 11B) among different groups in the GSE18816 dataset. Similarly, in the GSE41300 dataset, the expression levels of the feature genes PHGDH (*p* = 0.71) (Figure 11C) and NLRP12 (*p* = 0.78) (Figure 11D) across different groups are not statistically significant. In the dataset involving laboratory-cultured macrophages infected with viruses, the GSE141236 dataset shows no significant difference in the expression levels of the feature genes PHGDH (*p* = 0.10) (Figure 11E) and NLRP12 (*p* = 0.08) (Figure 11F) between three macrophages infected with HCMV and three uninfected control macrophages. Likewise, in the GSE162261 dataset, the expression levels of the feature genes PHGDH (*p* = 0.10) (Figure 11G) and NLRP12 (*p* = 1.00) (Figure 11H) between three samples infected with IAV and three uninfected control samples are not statistically significant. Based on these results, we tentatively suggest that the feature genes PHGDH and NLRP12 may serve as a potential combination of feature genes for SFTSV infection.

## 4. Discussion

In recent years, there has been some progress in the research on SFTS. However, several challenges still exist. Firstly, due to the requirement of conducting research on SFTS virus in Biosafety Level 2 or higher laboratories, sample collection and experimental operations are limited. This makes it difficult to obtain an adequate number of patient samples and conduct thorough experimental analysis. Furthermore, SFTS is a newly emerging disease, and our understanding of its pathogenic mechanisms and immunological characteristics is still relatively limited, which adds complexity to the research. However, THP-1 cells induced by PMA have significant value and significance as cellular materials for studying SFTS. THP-1 cells are a human monocytic cell line that can be differentiated into macrophage-like cells and exhibit similar functions and phenotypes to human macrophages. By using PMA induction, THP-1 cells can simulate mature macrophages and better mimic actual infection conditions [31]. Therefore, utilizing PMA-induced THP-1 cells for SFTS-related research contributes to a deeper understanding of the disease’s pathogenesis and immune responses.

Moreover, transcriptomic studies play a crucial role in revealing disease mechanisms and identifying potential biomarkers. For the transcriptomic study of SFTS, we established infected groups at different time points (0.5 h, 2 h, 8 h, 24 h, 48 h) and control groups. Through transcriptomic analysis of THP-1 cells infected with SFTSV after PMA induction, we obtained detailed gene expression profiles at each time point. This transcriptomic study helps us understand the gene regulatory networks and signaling pathway changes during SFTS virus infection as well as discover potential targets related to disease progression. In addition, our research employed various bioinformatics analysis methods for the first time, such as sample expression profile correlation analysis, cluster heatmap analysis, and cluster dendrogram analysis. All of these analyses suggest that SFTSV-infected THP-1 cells can be classified into four stages: 0.5–2 h, 8 h, 24 h, and 48 h. The significance of this clustering result lies in revealing the dynamic gene expression changes during SFTSV infection, which further aids our understanding of the virus–host cell interactions and their impact on the host’s immune response. Through in-depth transcriptomic analysis and clustering results, we can gain a more comprehensive understanding of the pathogenic mechanisms and pathological processes of SFTS, providing vital clues for the discovery of new therapeutic targets and preventive measures.

During the analysis and screening of expression profiles at different time points of THP-1 cells infected with SFTSV using bioinformatics analysis methods, we identified two crucial genes: PHGDH and NLRP12. These genes not only displayed similar changes in expression levels during SFTSV infection but also exhibited significant representativeness at each time point of infection. Therefore, through LASSO/Cox regression analysis, we successfully enriched these two important genes across all time periods. Furthermore, external validation analysis of the expression profiles from the public dataset GSE144358 revealed that the expression levels of PHGDH and NLRP12 remained statistically significant (*p* < 0.05). Particularly, PHGDH showed a remarkably high correlation with the severity of SFTS disease. These findings suggest that the PHGDH and NLRP12 genes hold promise as potential characteristic genes for monitoring the progression of SFTS disease.

Further review of the literature uncovers the extensive research significance and value of PHGDH and NLRP12 in virus-infection-related studies. Firstly, PHGDH encodes phosphoglycerate dehydrogenase, which plays a role in serine synthesis within cellular metabolic pathways. Studies have demonstrated that serine metabolism plays a pivotal role in viral infections, influencing viral replication, spread, and modulation of immune responses [32]. Consequently, investigating the regulatory mechanisms of PHGDH during viral infection can shed light on its relationship with viral replication and immune response regulation and provide a deeper understanding of the pathogenesis of viral infections. Additionally, NLRP12 belongs to the NOD-like receptor family and serves as an essential inflammatory regulatory protein. Recent research has indicated that NLRP12 exerts negative regulation on inflammation during viral infections and impacts immune cell apoptosis [33,34]. The regulation of inflammatory responses and the delicate balance between immune cell survival and apoptosis are critical factors in the development and outcome of viral infections. Hence, studying the functionality and regulatory mechanisms of NLRP12 can contribute to a comprehensive comprehension of the regulatory network governing immune responses and the mechanisms underlying the interaction between viruses and host immunity.

At the cellular level, various techniques can be employed to investigate PHGDH and NLRP12. For instance, gene knockout, overexpression, or silencing methods can be utilized to explore their functions during viral infections. Moreover, examining protein interactions and signaling pathways can unveil the intricate interplay and regulatory mechanisms involving PHGDH and NLRP12 within cells. These studies provide valuable insights into the molecular mechanisms of viral infections and cellular biology processes, paving the way for potential therapeutic targets and intervention strategies. Furthermore, at the clinical population level, studying the significance and value of PHGDH and NLRP12 in viral infections becomes even more imperative. Investigating gene polymorphisms and expression levels in large-scale populations enables the evaluation of the association between PHGDH, NLRP12, susceptibility to viral infections, disease severity, and treatment response. This approach facilitates the prediction of individual susceptibility to viral infections and the identification of novel biomarkers and supports personalized treatment and intervention strategies. In conclusion, PHGDH and NLRP12 play significant roles in virus-infection-related research. By delving deeper into their functionalities, regulatory mechanisms, and their relationship with viral infections at both the cellular and clinical population levels, we can enhance our understanding of viral pathogenesis and immune regulation processes and identify potential therapeutic targets and preventive strategies.

This study exhibits several notable strengths and innovations. Firstly, it demonstrates meticulous experimental design and handling of multiple time points of SFTSV infection, coupled with extensive comparative analysis of the corresponding expression profiles. The successful application of clustering analysis effectively stratifies samples and unveils distinct data patterns. Additionally, this study employs sophisticated techniques such as soft clustering analysis and LASSO/Cox machine learning methods, enabling comprehensive assessment of expression profile dynamics across different infection time points. The identification of two critical genes, namely PHGDH and NLRP12, which were further validated in an independent clinical dataset, underscores the robustness of the findings. In addition, we performed a nuanced analysis of the specific expression profiles of the feature genes in four supplementary external datasets (Accession IDs: GSE18816, GSE41300, GSE141236, GSE162261). The outcomes of this analysis lend credence to the possibility that both the PHGDH and NLRP12 genes could potentially function as distinctive feature genes.

Nevertheless, it is essential to acknowledge certain limitations within this study. Firstly, there is a need to enhance the accuracy and stability of the employed data analysis methods. Furthermore, incorporating additional independent validation datasets would provide further validation and bolster the reliability of the findings. A deeper investigation into the precise functional mechanisms of the identified genes warrants attention. Expanding the research scope to encompass a broader array of relevant genes and pathways would yield a more comprehensive understanding of the subject matter. It has also been reported that PBL-1, bearing a similar immunophenotype to SFTSV target cells in fatal SFTS, serves as a potential in vitro model for human SFTSV infection [35]. Consequently, we plan to further expand and supplement relevant research content in subsequent studies. Addressing these limitations would significantly elevate the scientific rigor and translational potential of this study, ultimately contributing to advancements in SFTSV infection research and clinical practice.

## 5. Conclusions

This study implemented experimental designs and procedures at multiple time points of SFTSV infection, followed by extensive comparisons and analyses of expression profiles at these different time points. Through cluster analysis, effective sample classification was achieved, revealing clear structural patterns within the data. Furthermore, soft clustering analysis and LASSO/Cox machine learning methods were employed to successfully analyze the changes in expression profiles across various infection time points, leading to the identification of two important genes, PHGDH and NLRP12. And these findings were further validated in an external clinical dataset, reinforcing their significance and reliability.

## Figures and Tables

**Figure 1 viruses-16-00059-f001:**
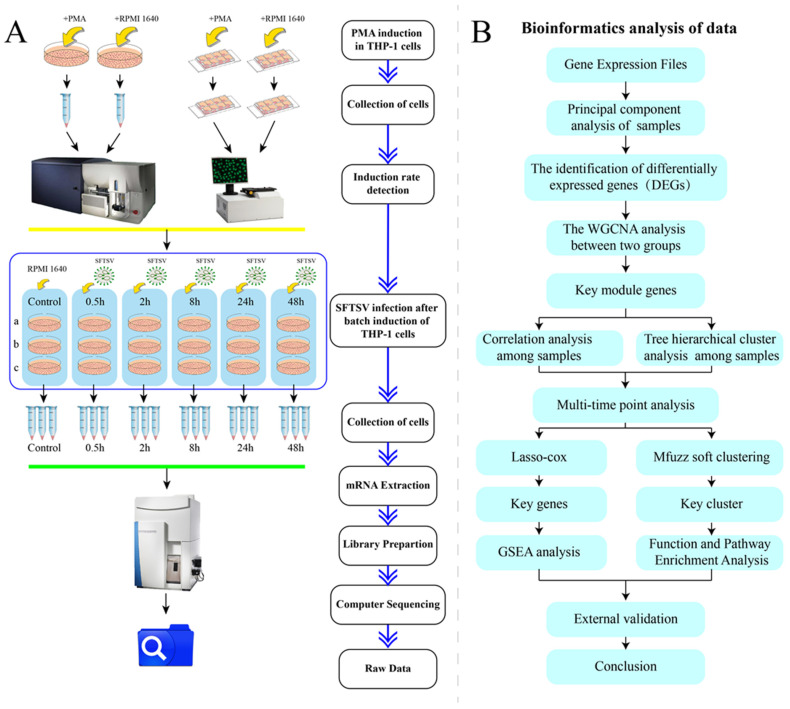
Illustrates the experimental design flowchart and data analysis process. (**A**) The preparation of cell samples for high-throughput RNA sequencing is depicted. (**B**) A bioinformatics analysis protocol is outlined.

**Figure 2 viruses-16-00059-f002:**
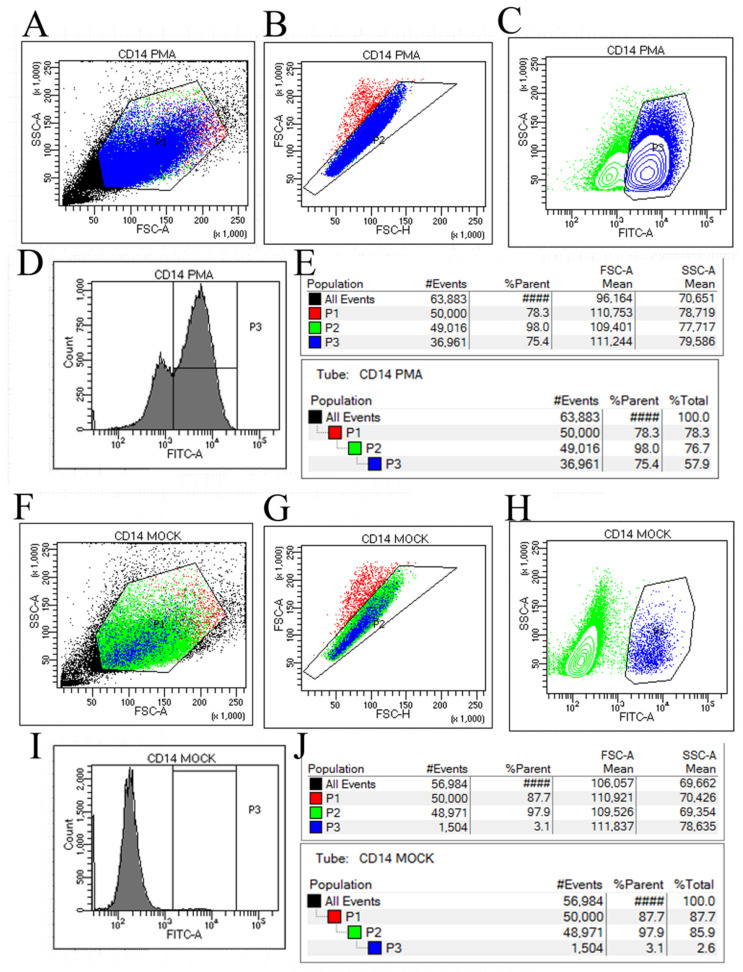
The results of THP-1 cell sorting using a flow cytometer with CD14-FITC labeling for cell sorting. (**A**–**C**) represent the gating strategy employed during the sorting of PMA-induced THP-1 cells, labeled as Gate 1 to Gate 3. (**D**) displays a peak map illustrating the cellular composition and distribution resulting from the sorting of PMA-induced THP-1 cells. (**E**) presents the percentage composition of FITC-positive cells obtained from the sorting of PMA-induced THP-1 cells. Similarly, (**F**–**H**) depict the gating strategy utilized during the sorting of THP-1 cells in the control group. (**I**) exhibits a peak map illustrating the cellular composition and distribution resulting from the sorting of THP-1 cells in the control group. Finally, (**J**) provides the percentage composition of FITC-positive cells obtained from the sorting of THP-1 cells in the control group.

**Figure 3 viruses-16-00059-f003:**
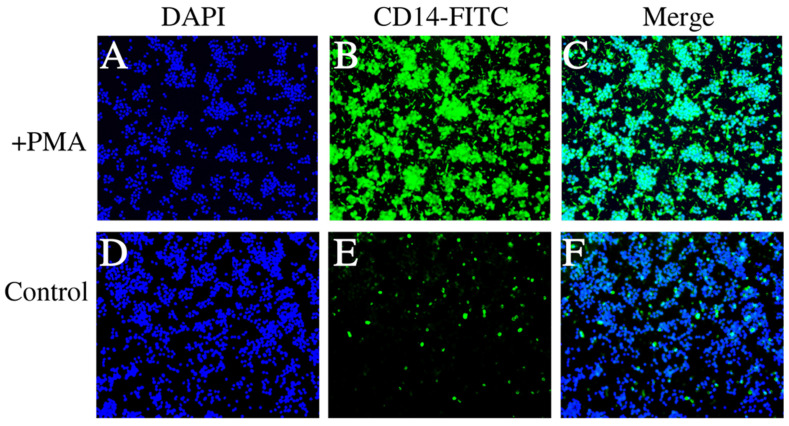
Fluorescent labeling of THP-1 cells. (**A**–**C**) represent the labeling of dTHP-1 cells using markers DAPI, FITC, and Merge. (**D**–**F**) exhibit the labeling of control THP-1 cells using markers DAPI, FITC, and Merge, while the surrounding THP-1 cells showed no fluorescence signal.

**Figure 4 viruses-16-00059-f004:**
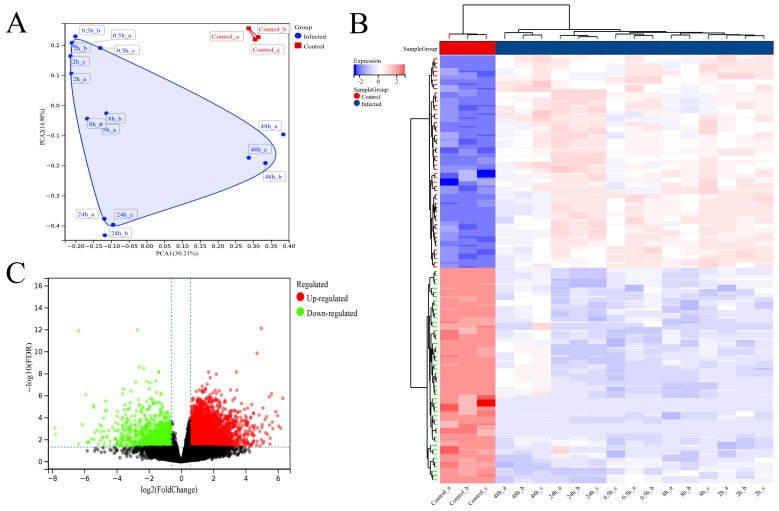
Differential analysis between SFTSV-infected THP-1 cells and the control group. (**A**) Principal component analysis (PCA) results of cell samples. (**B**) Heatmap analysis results depicting the differential clustering between SFTSV-infected THP-1 cells and the control group. (**C**) Volcano plot analysis results showing the differential gene expression between SFTSV-infected THP-1 cells and the control group.

**Figure 5 viruses-16-00059-f005:**
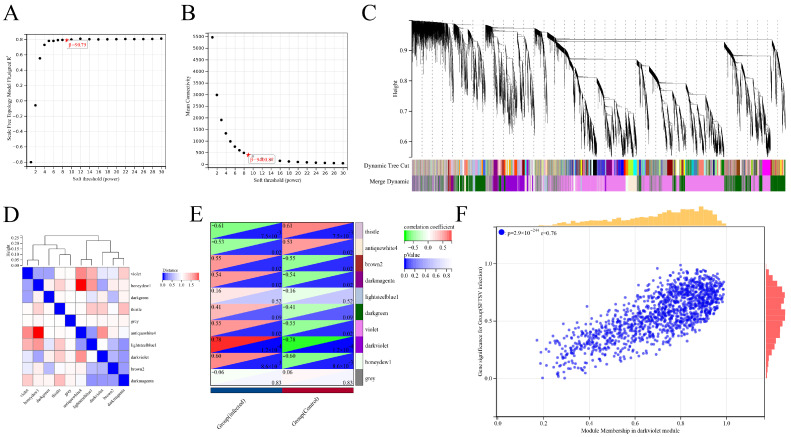
Analysis of differentially expressed gene (DEG) dataset using weighted gene co-expression network analysis (WGCNA) and identification of candidate key genes. (**A**) Results of selecting the soft threshold power in WGCNA. (**B**) Average connectivity results obtained from WGCNA. (**C**) Clustering tree results generated by WGCNA. (**D**) Correlation analysis results illustrating the relationship between WGCNA clustering modules. (**E**) Correlation analysis results depicting the association between WGCNA clustering modules and differential grouping. (**F**) Results of correlation analysis showing the module membership in the darkviolet module and gene significance for the SFTSV-infected group.

**Figure 6 viruses-16-00059-f006:**
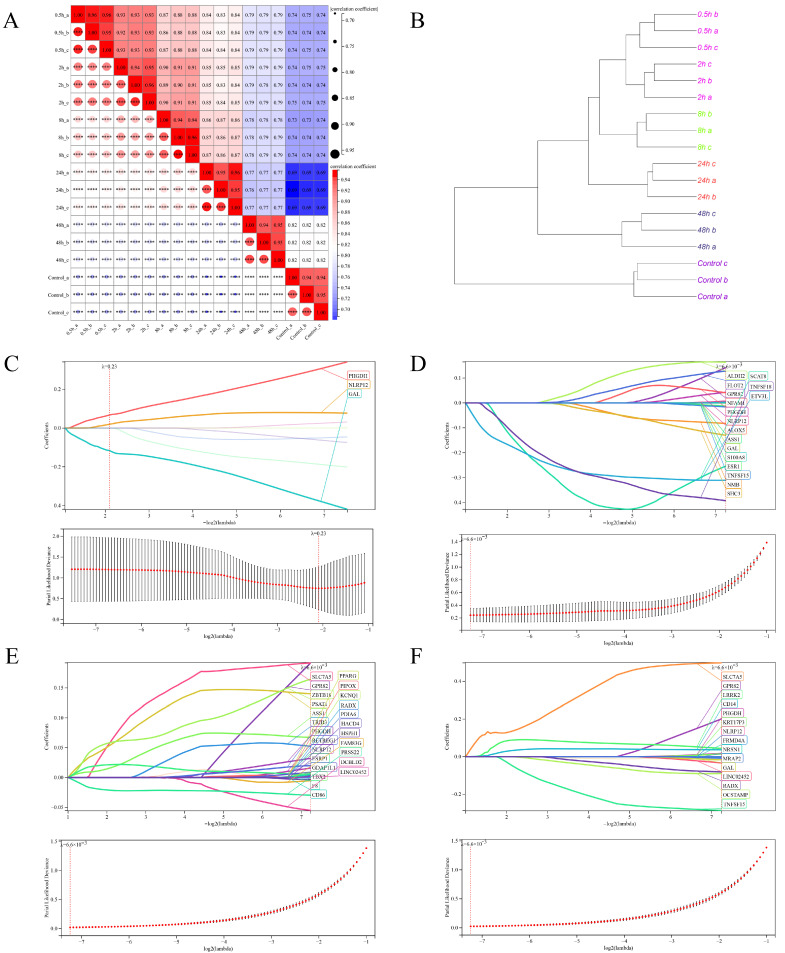
Clustering analysis of all samples and identification of key feature genes using LASSO/Cox analysis. (**A**) Heatmap depicting the correlation analysis between samples. (**B**) Dendrogram displaying the hierarchical clustering results based on gene expression profiles among samples. The sample groups at 0.5–2 h, 8 h, 24 h, and 48 h were compared to the control group using the LASSO/Cox regression model. This analysis identified characteristic genes with a fixed lambda value determined through cross-validation for each time point: (**C**) 0.5–2 h, (**D**) 8 h, (**E**) 24 h, and (**F**) 48h.

**Figure 7 viruses-16-00059-f007:**
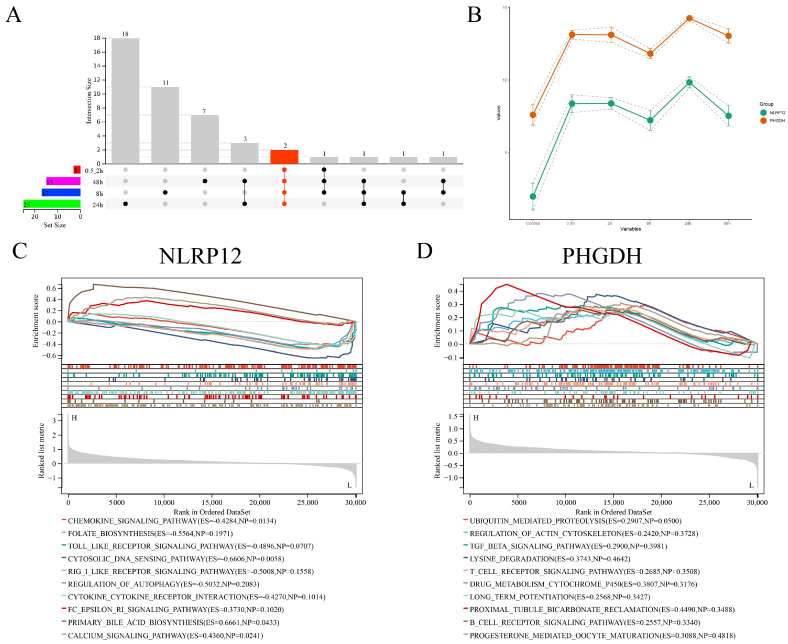
UpSetR intersection results and GSEA analysis of the selected feature genes across different groups. (**A**) UpSetR intersection plot displaying the intersection results of the selected feature genes from each group. (**B**) Expression trends and 95% confidence interval ranges of two key genes within the UpSetR intersection results at each time point. (**C**) GSEA analysis results for the key gene NLRP12. (**D**) GSEA analysis results for the key gene PHGDH.

**Figure 8 viruses-16-00059-f008:**
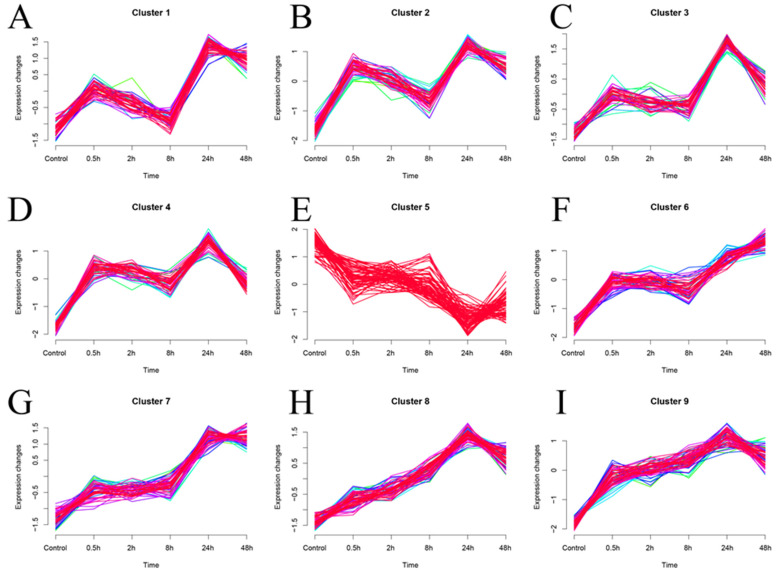
Fuzzy clustering (MUfzz) analysis of the expression profiles of 515 hub genes obtained from WGCNA analysis at different time points of SFTSV infection. (**A**–**I**) Presentation of 9 MUfzz results on 515 hub genes. Within the same cluster, the presence of lines in various colors reflects the sample’s fuzzy membership to multiple clusters.

**Figure 9 viruses-16-00059-f009:**
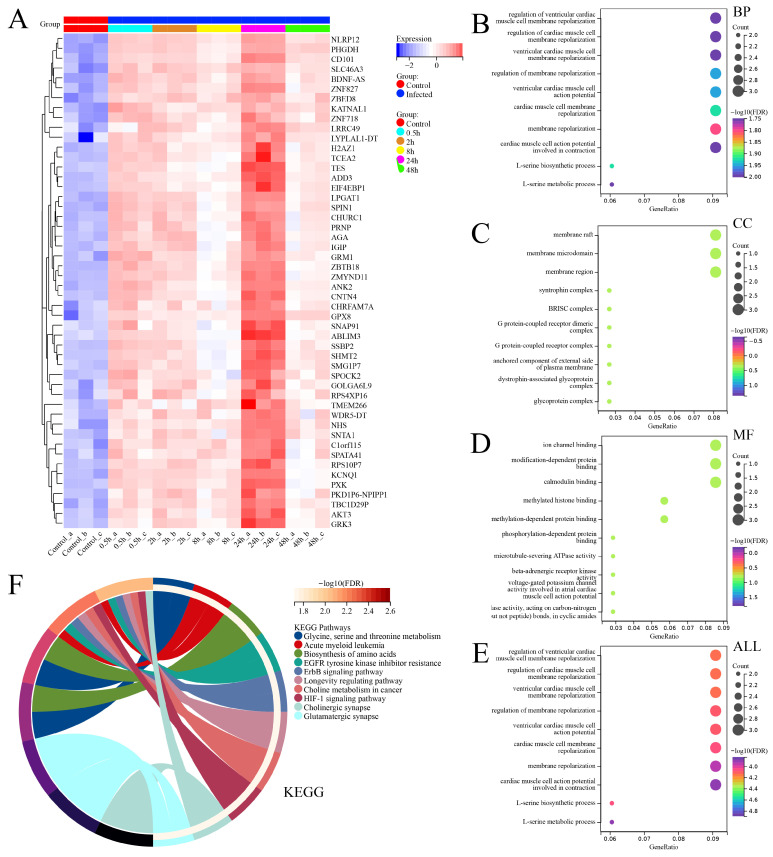
Expression analysis and functional enrichment analysis of the gene set obtained from fuzzy clustering (MUfzz) analysis, specifically cluster 4. (**A**) Heatmap depicting the expression levels of 50 genes within the cluster 4 gene set. (**B**) GO enrichment results for biological process (BP). (**C**) GO enrichment results for cellular component (CC). (**D**) GO enrichment results for molecular function (MF). (**E**) GO enrichment results for all GO categories combined (ALL). (**F**) KEGG pathway enrichment results.

**Figure 10 viruses-16-00059-f010:**
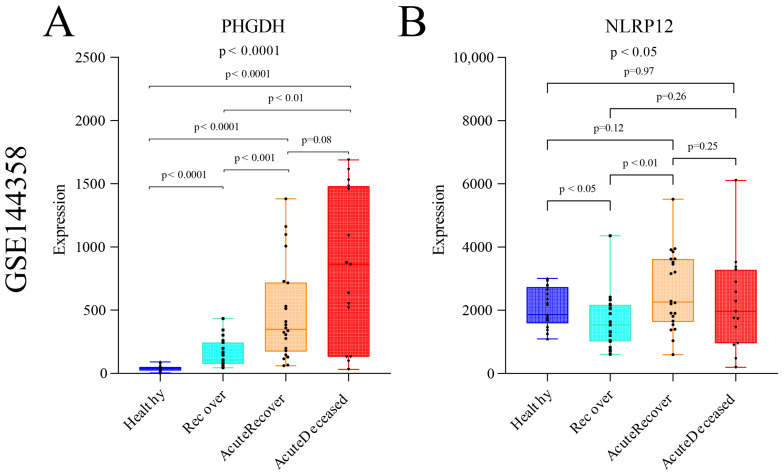
Expression analysis of gene PHGDH and gene NLRP12 in SFTS patient samples with different disease severity levels from GSE144358. (**A**) Expression changes of gene PHGDH in SFTS patient samples with different disease severity levels. (**B**) Expression changes of gene NLRP12 in SFTS patient samples with different disease severity levels.

**Figure 11 viruses-16-00059-f011:**
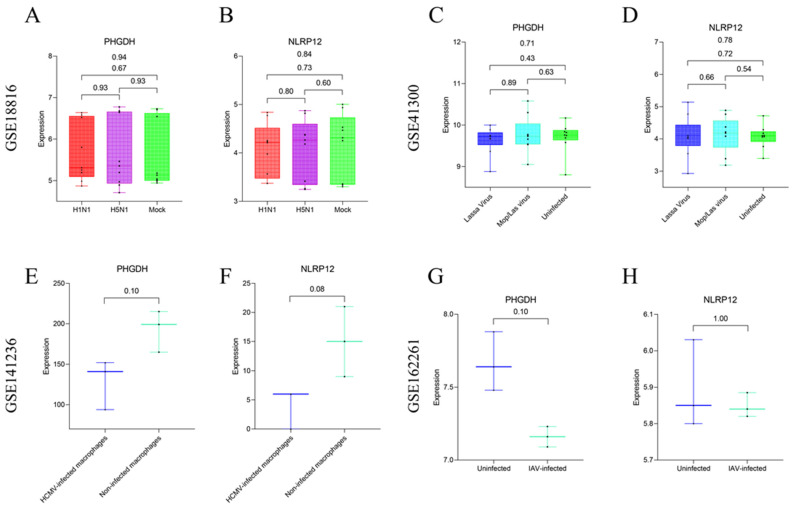
Evaluation of the expression of target feature genes in clinical samples and laboratory-cultured macrophages using four external validation datasets. (**A**,**B**) Expression levels of the PHGDH and NLRP12 genes in different groups (H1N1 infection group, H5N1 infection group, mock group) within the GSE18816 dataset. (**C**,**D**) Expression levels of the PHGDH and NLRP12 genes in different groups (Lassa virus infection group, Mop/Las virus infection group, uninfected control group) within the GSE41300 dataset. (**E**,**F**) Expression levels of the PHGDH and NLRP12 genes in different groups (macrophages infected with HCMV and uninfected control macrophages) within the GSE141236 dataset. (**G**,**H**) Expression levels of the PHGDH and NLRP12 genes in different groups (IAV infection group and uninfected control group) within the GSE162261 dataset.

**Table 1 viruses-16-00059-t001:** Results of data quality control for transcriptome sequencing.

Sample	Raw Reads	Clean Reads	Clean Bases	Error (%)	Q20 (%)	Q30 (%)	GC (%)
Con_a	51351534	49380376	7.39 G	0.04	97.56	93.43	47.29
Con_b	48700228	46605374	6.97 G	0.04	96.98	92.03	46.88
Con_c	48772582	46967334	7.03 G	0.04	97.15	92.17	47.08
0.5h_a	57442976	55542550	8.3 G	0.04	97.87	94.31	46.16
0.5h_b	55893148	54049126	8.08 G	0.04	97.6	93.54	46.08
0.5h_c	48629620	47049176	7.04 G	0.04	97.39	92.87	46.2
2h_a	56381612	54489560	8.16 G	0.04	97.56	93.4	46.79
2h_b	55676058	53751606	8.04 G	0.04	97.51	93.26	46.71
2h_c	55803544	53802804	8.05 G	0.04	97.46	93.18	46.67
8h_a	50837064	49222058	7.34 G	0.04	97.89	94.25	47.43
8h_b	47859920	46116568	6.9 G	0.04	97.46	93.2	47.03
8h_c	54598234	52678942	7.88 G	0.04	97.4	92.93	47.07
24h_a	50530122	48702430	7.29 G	0.04	97.17	92.27	47.38
24h_b	50848524	48895934	7.32 G	0.04	97.59	93.54	47.09
24h_c	47937234	46222114	6.92 G	0.04	97.35	92.78	47.15
48h_a	47382892	45919646	6.87 G	0.04	97.85	94.12	47.41
48h_b	53639272	51868716	7.76 G	0.04	97.65	93.62	47.16
48h_c	58370484	56457460	8.44 G	0.04	97.64	93.49	47.51

## Data Availability

The original contributions presented in the study are included in the article and Appendix A. Further inquiries can be directed to the corresponding author or the first author.

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
