# Peer review of "Time-Course Transcriptome Analysis Reveals Distinct Phases and Identifies Two Key Genes during Severe Fever with Thrombocytopenia Syndrome Virus Infection in PMA-Induced THP-1 Cells"

_viruses, 2023, doi:10.3390/v16010059_

Round 1

Reviewer 1 Report

Comments and Suggestions for Authors

In this study, experimental designs and procedures at multiple time points of SFTSV infection were performed using PMA-induced THP-1 cells, followed by extensive comparison and analysis of expression profiles at these different time points. During the analysis and screening of expression profiles at different time points of THP-1 cells infected with SFTSV using bioinformatics analysis methods, the authors identified two crucial genes: PHGDH and NLRP12. These experimental methods and new discoveries about important genes in SFTS infection will be important for future SFTS research. I think the findings of this study are relevant to the scope of the journal and will be of interest to its readership. However, the several points described below may be better to be re-considered.

Major comments: 

1.      It has been reported that the majority of SFTSV-infected cells were B cell–lineage lymphocytes, and a human plasmablastic lymphoma cell line, PBL-1, was susceptible to SFTSV propagation and had a similar immunophenotype to that of target cells of SFTSV in fatal SFTS (Suzuki T, et al. 2020, J Clin Invest). While it is important to analyze SFTSV-infected macrophages to assess the host immune response to SFTSV, the significance and limitations of using macrophages rather than B-cell lymphocytes would be better discussed in the introduction or discussion section.

2.      Lines 398–401: The authors mention that viral infections can have direct effects on cardiac tissue, and the dysregulation of membrane repolarization and ion channel activities in cardiac muscle cells can contribute to cardiac dysfunction. Autopsy cases of SFTS patients have been reported with and without SFTSV infection of myocardial cells (Saijo M. 2018, J Infect Chemother). The reviewer does not understand whether the authors are assuming that SFTS infection of myocardial cells causes dysregulation of membrane repolarization or that SFTS-infected macrophages affect the myocardium. Based on the results of this study, is it appropriate to mention the direct effects of viral infection on cardiac tissue?

Minor comment:

Lines 357–362: Similar phrases are used repeatedly in this sentence, making it difficult to understand the content.

Author Response

Dear reviewers,

Thank you for your letter and the reviewers’ comments concerning our manuscript entitled “Time-course transcriptome analysis reveals distinct phases and identifies two key genes during SFTSV infection in PMA-induced THP-1 cells” (viruses-2717180). We are all honored to get your valuable opinions. Those comments are valuable and very helpful. We have verified and modified the article based on your suggestions. Additionally, following your recommendation, we also conducted a more specific and comprehensive expression profile analysis of the target genes in four external datasets (Accession IDs: GSE18816, GSE41300, GSE141236, GSE162261).Based on the instructions provided in your letter, we uploaded the file of the revised manuscript.

I wish you all the best.

All author respects

December 16, 2023

Reviewer 2 Report

Comments and Suggestions for Authors

Authors suggested the experimental designs, extensive comparisons and analyses of expression profiles at different time points of SFTSV infection using THP-1 cells. These approaches seem to be very useful to detect the specific genes, proteins or markers.

Author Response

(The authors gave the same response as above.)

Reviewer 3 Report

Comments and Suggestions for Authors

- In line 37, Severe Fever with Thrombocytopenia Syndrome Virus (SFTSV) is a highly pathogenic pathogen belonging to the Bunyavirales order.

: SFTSV's official name is Dabie bandavirus. Could authors put the official name?

- Published papers showed that levels of both IL-10 and IL-6 were significantly elevated, the level of transforming growth factor-β (TGF-β) was significantly decreased and IL-10 was elevated earlier than IL-6 in fatal SFTS patients, and inhibition of IL-10 signaling decreased the production of IL-6 and elevated that of TGF-β through SFTSV infected THP-1-derived macrophages

(ref 1. Fatal outcome of severe fever with thrombocytopenia syndrome (SFTS) and severe and critical COVID-19 is associated with the hyperproduction of IL-10 and IL-6 and the low production of TGF-β. J Med Virol. 2023 Jul;95(7):e28894. 

ref. 2. IL-6 and IL-10 Levels, Rather Than Viral Load and Neutralizing Antibody Titers, Determine the Fate of Patients With Severe Fever With Thrombocytopenia Syndrome Virus Infection in South Korea. Front Immunol. 2021 Aug 17;12:711847). 

Therefore, could authors show the signal pathway data (or results) about IL-10, IL-6, and TGF-β in this study?

Comments on the Quality of English Language

The English Language is satisfactory. 

Author Response

(The authors gave the same response as above.)

Round 2

Reviewer 2 Report

Comments and Suggestions for Authors

Authors revised well. 

Author Response

Dear reviewers,

Thank you for your letter and the thoughtful comments from the reviewers regarding our manuscript titled "Time-course transcriptome analysis reveals distinct phases and identifies two key genes during SFTSV infection in PMA-induced THP-1 cells" (viruses-2717180). We feel honored to receive your valuable opinions, which we find to be insightful and constructive. Your comments have proven to be exceptionally helpful, and we have taken the time to carefully review and incorporate the necessary modifications into the article based on your suggestions. In accordance with the guidance provided in your letter, we have submitted the revised manuscript for your consideration.

I wish you all the best.

All author respects

December 24, 2023
